# Narratives on the Current Medical Situation in Japan According to Highly Specialized Foreign Professionals

**DOI:** 10.3390/healthcare10091694

**Published:** 2022-09-05

**Authors:** Tomoari Mori, Yoko Deasy, Eri Kanemoto, Eisuke Nakazawa, Akira Akabayashi

**Affiliations:** 1Health Center, Okinawa Institute of Science and Technology Graduate University (OIST), Okinawa 904-0495, Japan; 2Department of General Medicine, University of Occupational & Environmental Health, Japan (UOEH), Fukuoka 807-8555, Japan; 3Department of Biomedical Ethics, University of Tokyo Faculty of Medicine, Tokyo 113-0033, Japan; 4Division of Medical Ethics, New York University School of Medicine, New York, NY 10016, USA

**Keywords:** foreign residents in Japan, medical issues, qualitative research, language support, intercultural communication

## Abstract

In order to understand the difficulties faced by highly skilled foreign professionals when dealing with the Japanese healthcare system and to identify the support they require therein, university health center staff members of the Okinawa Institute of Science and Technology Graduate University conducted semi-structured interviews with faculty, staff, and students from the Institute. Data from the interviews were analyzed by subject matter analysis using a narrative-oriented approach. In total, 13 participants were interviewed, and five themes and 15 subthemes were generated from the 40 codes extracted. Although participants considered themselves to be accepting of other cultures and made little mention of the need for cultural and religious considerations that previous studies have identified as important, they reported that their experiences receiving healthcare in Japan were fraught with many difficulties. They felt that the capacity to communicate in Japanese was a prerequisite for receiving appropriate healthcare and that hospitals should assume the responsibility of providing language support. While they reported satisfaction with the easy and inexpensive access to advanced medical equipment and specialists in Japan, they also noted challenges in selecting medical institutions and departments, the flow and procedures in the hospital, and building open and direct relationships with doctors. In addition, based on the present study, people with chronic illnesses felt isolated from the community, worried about a lack of privacy, and wanted a primary care physician they could trust. In order to provide appropriate healthcare to foreigners, we require an accurate understanding of their needs, how to address these comprehensively and in a multifaceted manner, and how the communication responsibilities should be shared among the involved parties (i.e., foreign care recipients and Japanese medical professionals).

## 1. Introduction

In recent years, with the rapid increase in the number of foreign tourists and foreign residents, so-called foreigner-related medical problems have emerged in Japan. Major problems include a shortage of medical interpreters, unpaid medical bills, and the introduction of imported infectious diseases; the government has already taken various steps to address many of these [1]. In addition, third-party certification systems such as the Japanese Medical Institution Certification Program (JMIP) [2], certified medical interpreters, and Japanese international nurses [3] have been initiated, and the healthcare delivery system for foreigners is rapidly improving. This trend was expected to accelerate and continue in the lead-up to the 2020 Tokyo Olympics.

An evaluation of the quality of healthcare for foreigners is required. In addition to addressing the problems faced by Japanese medical professionals, it is also necessary to ensure that appropriate healthcare can be offered to the many foreigners with diverse backgrounds. To this end, evaluation of the healthcare provided to foreigners must be conducted, specifically from the recipients’ perspectives. In-depth research on the adaptation process of ethnic groups to different cultures has been conducted focusing on minorities, especially in the Western world, where ethnic migration has been historically repeated [4,5,6,7,8]. These studies have established a field of cross-cultural adaptation and understanding in healthcare settings [9,10,11,12,13,14,15]. However, Japan is an island nation comprising a single ethnic group, speaking almost exclusively Japanese. Although English education has long been compulsory, the country has not yet become multi-ethnic or multilingual. In recent years, Japan has seen rapid globalization, highlighting the need to secure an efficient and young workforce to overcome a super-aging society. The Japanese government has commenced a policy of emphasizing the importance of the tourism industry and attracting white-collar workers along with blue-collar workers. Consequently, the number of foreign tourists and foreign residents in Japan has been increasing rapidly. However, the system for accepting foreign nationals is still in its nascent stage, and the medical field has limited experience in accepting foreign personnel in various sectors. In addition, there is still little research from the perspective of cross-cultural understanding and response in this regard. Some studies have been conducted to expand our knowledge of the experiences and difficulties associated with providing services to foreign patients from the perspective of healthcare professionals, especially nurses [16,17]. Unfortunately, studies focusing on healthcare evaluation from the patient’s perspective are limited, and “foreigners living in Japan face language barriers [18]”. Accordingly, few studies have assessed these perspectives using a narrative approach.

In recent years, a new category of foreigners labeled “advanced foreign human resources” has been promoted as a policy measure [19], and Japan’s universal health insurance system is reportedly attractive to such foreigners [20]. Their satisfaction may serve as a useful barometer against which to evaluate the international acceptance of and satisfaction with Japanese healthcare. However, perceptions of the actual experience of receiving healthcare remain unclear.

Against this backdrop, in the present study, we surveyed the foreign community at the Okinawa Institute of Science and Technology Graduate University (OIST) [21] to identify difficulties related to Japanese healthcare faced by university faculty, staff, and students, all of whom are highly specialized foreign professionals, and to understand the form of support they desire.

OIST was established to promote the formation of a world-class research center by inviting outstanding researchers from Japan and abroad to conduct high-quality research.

Over half of OIST faculty and students are recruited from overseas, and all teaching and research is conducted in English [22]. This environment at OIST is considered appropriate for conducting this research.

The OIST Health Center staff who interviewed for this study have provided medical consultation support to foreign resident faculty, staff, students, and accompanying family members since the opening of the university. Although we have dealt with many problems and complaints from both foreigners and Japanese healthcare providers regarding medical visits, these problems required merely practical and technical solutions. Until now, there has been no opportunity to deeply investigate or consider the cross-cultural communication and understanding that lies behind problems in daily work. Therefore, we aimed to extract stories, including personal feelings and opinions regarding foreigners’ perception of difficulties in visiting Japanese medical facilities and their experiences of overcoming these difficulties. The OIST health center staff, who had been involved in supporting foreign residents in their medical visits, listened to their stories and tried to understand them. The outcome of the study was to gain a foothold for the inclusion of foreigners in local healthcare and medical support professions in Japan.

## 2. Participants and Methods

### 2.1. Participants

Participants were faculty, staff, and students enrolled at OIST. In order to obtain a variety of opinions from the different backgrounds of the participants and to minimize the possibility of background bias, participants were recruited using a purposive sampling approach. The nature and purpose of the study were explained in writing and orally by the researcher, and written consent was obtained from potential participants if they agreed to participate in the study. This study was conducted after ethical review and approval by the OIST Human Subjects Research Review Committee (review number: HSR-2020-017).

Interviews were conducted with 13 participants (Table 1). As no new relevant codes were extracted after the 13th participant’s interview, the collected content was considered saturated and recruitment was terminated at that point. Only one interview was conducted with each participant.

### 2.2. Methods

We used a narrative-oriented approach. Interviews were conducted by an occupational physician (T.M.: M.D., Ph.D., MPH, male, aged 50–60), a university health nurse (D.Y.: RN, PHN, female, aged 30–40), and an administrative employee (E.K., female, aged 50–60) from August 2020 to May 2021. The main interviewer (D.Y. or E.K.) asked the main questions, while the sub-interviewer (T.M.) was responsible for reviewing and organizing the interview process and asking questions for content review. All interviews were conducted in English in a private room on campus, either in person or via Zoom. The interviews were semi-structured and followed an interview guide (Table 2).

Questionnaires were developed from previous research studies and discussions based on researcher experience. According to researcher experience, health center staff members have cared for foreigners since the university opened, handling complaints and other concerns. At the time of the present study, they had not heard about the cultural and religious issues mentioned by previous studies; rather, the general impression was that many complaints pertained to convenience, or family matters. Specifically, medical visits could be stressful, especially for children.

We collected data on the basic attributes of the participants, the pros and cons of Japanese healthcare relative to that in their countries of origin, examples of cases in which they or their family members had received healthcare, problems they had faced in receiving healthcare, and the support they felt they needed.

Each interview lasted 1 to 1.5 h, and the content was recorded with interviewee consent. Recorded content was transcribed and analyzed using NVivo (Release 1.6.1, QSR International, Burlington, MA, USA) using the thematic analysis method [23]. The analysis procedure was as follows.

Verbatim transcripts were divided into text segments by semantic content in NVivo by TM, and each segment with content relevant to the study was assigned a descriptive code.Following completion of all interviews, all researchers reviewed the content, selected the next potential participant, reviewed the interview guide and questions, and revised the content as needed.The derived codes were then repeatedly compared and grouped according to semantic content to extract each theme and subtheme (Table 2). From this point on, all researchers discussed and analyzed the data.By examining the contextual meaning of each code and examples of statements, a diagram of the relationship between categories and subcategories (causal and oppositional) was created (Figure 1).Based on the above analysis, we reconstructed the participant comments as a story.Since we were particularly interested in the gains and losses of language support methods, we separately extracted the relevant text segments and organized what was said about their characteristics and advantages and disadvantages (Table 3).Finally, Figure 1, Table 2, Table 3, and the reconstructed story that was created were reviewed by three representatives from among the participants to confirm the validity of the deliverables.

Medical facility environments near the OIST:500 m: Clinic (interpreter software available for foreigners to use)3.5 km: Clinic (for Japanese, but not restricted to foreigners)6.5 km: Clinics of Psychiatry, Internal Medicine, and Otolaryngology18 km: General hospital (2.5 tier)17 km: General hospital (tertiary)25 km: General hospital (certified as Japan Medical Service Accreditation for International Patients)17.7 km: Radiotherapy and medical checkup clinic (outsourced medical examinations and acceptance of foreign nationals are also advocated)

Roughly 20 other hospitals were also accessible to those in the southern part of the province, mainly because it is a small province.

On-campus medical facilities:

OIST Clinic: Located on the university grounds 

Health Center: Open weekdays 9 a.m. to 5 p.m. (over-the-counter prescriptions can be filled); clinic is open by appointment only for half a day in the morning or afternoon (a possible change in the future), and first aid for sudden illnesses or emergencies is available on all days. Since health insurance is not available to cover treatment, employees pay 30% of the cost (the same amount as regular national insurance in Japan), and student services are free. The rest of the costs are supported by the university.

The amount given as a lost opportunity cost for an hour-long interview was 1000 yen (under 10 US dollars), which is roughly equivalent to the hourly minimum wage in Japan. The researcher intended that the gift card would show appreciation for participation but not be a substantial incentive enough to participate.

## 3. Results

Each theme and subtheme, as well as their constituent codes, were grouped and organized (Table 3). Interrelationships among the codes and subcodes are shown in Figure 1. The five themes and fifteen subthemes generated represent the subjective experiences of the participants and could be organized along a time axis. The conflict axis and causal arrows in Figure 1 are thought to represent not only the participants’ adaptation process but also their internal conflict and resolution process. The following sections describe the specific relationship between themes (a)–(e) and subthemes. Participant statements corresponding to the codes of each theme are enclosed in parentheses and indicated in italics.

In theme (a), the following text was used to describe a receptive attitude toward different cultures, a flexible acceptance of Japanese culture, and willingness toward active adaptation with generous university support. *“I believe that one should adapt to the culture of the country in which one is staying. Nevertheless, in my daily life, I rarely have to enter the local Japanese community, and thanks to the various support provided by the university, I do not face any major problems”.*

In theme (b), i.e., difficulties in receiving healthcare, they faced the reality that it was difficult for them to receive healthcare on their own due to the language barrier and lack of knowledge about the Japanese healthcare system. This contradicted their ideal of cross-cultural adaptation as described in theme (a). They believed that language support was the responsibility of medical institutions as part of proper healthcare, but found that the average medical institution in Japan did not prioritize providing said support to foreigners. They expressed this in their own words. An example is as follows: *“However, when visiting a medical institution for illness or injury, I had to deal with the situation on my own, without much prior knowledge. Choosing a department or medical facility was difficult, and the procedures at the general hospital were complicated, difficult, and scary. No one spoke English except the doctors, and no one could help me in the hospital. It is very difficult to explain and negotiate in Japanese when a patient suddenly becomes seriously ill. I think language support is the hospital’s responsibility, but the hospital asked me to bring an interpreter because the doctor was going to give an important explanation. Furthermore, I was turned away by a psychiatrist who said that he could not accommodate foreigners”.*

Theme (c) describes the process of coping and adapting to Japanese healthcare through trial and error and self-help efforts, as follows: *“In emergencies, I was accompanied by family members, friends, colleagues, and occupational health staff who could interpret for me. Compared to my medical experience in my home country, I often felt uncomfortable and frustrated in my communication with the medical staff. I learned how to deal with such situations by asking questions and making suggestions whenever possible, giving up and accepting the situation, and seeking practical solutions such as transferring to another hospital. I tried to communicate without hesitation, using a mixture of Japanese and English, even if it was only one language, and using applications. It was a good experience. I am now able to go to the hospital by myself for regular checkups and simple medical conditions. I was finding ways to cope, if not perfect”*. Theme (c) can be seen as an internal process of trying to overcome the contradiction between the ideals of (a) and the reality of (b).

Theme (d) indicates that as foreigners become accustomed to the Japanese healthcare system, they are able to view its characteristics objectively, and while they acknowledge the good aspects as good, they are willing to understand and accept the aspects they find uncomfortable or inconvenient as characteristics of Japanese culture. One participant stated: *“Access to specialists and advanced medical equipment is very good. Patients are generally satisfied with the level of healthcare and cost burdens in Japan. Although patients do not feel much discrimination against foreigners, doctors do not like to be questioned and it is difficult to establish an open and direct relationship with them. I also feel that the whole process of seeing a doctor and prescribing a medicine is bureaucratic and inflexible, but I guess that is part of Japanese culture”*. It is important to note here that foreigners believe that there is little discrimination against them, i.e., they make a sharp distinction between cultural discrimination and the existence of barriers to healthcare.

Under theme (e), participants expressed frustration with their surroundings and their own needs. Others cited ambivalent feelings toward occupational health staff in the workplace and the need for primary care physicians to become integrated advocates. One person with a chronic or serious illness expressed concern about the lack of health privacy in their small community, and at the same time felt lonely that they could not confide in or discuss their health problems with anyone. One participant stated: *“The occupational health staff at the university is a good source of information, but I am concerned about my health information being known to my workplace, for example, through workplace health checkups. In my country, family physicians take the lead in solving problems organically. In my case, however, no one took on this responsibility and no one was involved in resolving my medical condition. I felt so alone that I even went to counseling”.* While the other themes pertain less to emotional issues and instead focus on the convenience and responsibility of accessing proper healthcare and practical ways to solve the problems, the concerns about the right to privacy and issues with stigma discussed in theme (e) were emotional. Their arguments were based on their cultural backgrounds and affirmed them, in contrast to theme (d), which was an attempt to understand objectively Japanese healthcare in a cross-cultural context.

The main points pertaining to language assistance are shown in Table 4. The participants felt that, at least at public hospitals and general hospitals, they should be able to expect appropriate medical interpreting services such as local and online interpreters, and further stated that depending on the information needed, the nature of the problem, the size of the medical institution, and the Japanese language ability level of the patient, access to multifaceted support, including information on websites, multilingual entry forms, bilingual tables, multilingual specialists, multifaceted support provided by appropriate sources, etc., should be attainable.

At the beginning of this study, the interviewers anticipated that the content that interviewees considered to be difficult at the time of their visit and their demands regarding respect for culture and religion would be largely determined by their cultural backgrounds. However, the interviewees’ experiences could be analyzed in terms of length of stay and stage of experience and organized as themes (a)–(e) of the story of the adjustment process shared by the interviewees. In the following sections, we further describe the relationship between the interviewee and interviewer, including the aspects of the individual narratives along the time axis that link each theme.

The main interviewers, public health nurses, and administrative staff had been engaged in supporting researchers for about 10 years, since the opening of the OIST. Public health nurses, in particular, had direct contact with patients, so they remembered most episodes of medical visits described by the interviewees from their own perspective. For the interviewees, sharing episodic experiences with the interviewer seemed to evoke a sense of wanting to be heard and understood, which in this study worked to their advantage in understanding their trials and tribulations. In cases where the health worker was a friend or family member of the interviewee, the administrative staff acted as the main interviewer to avoid any bias. The sub-interviewee, an industrial physician, had been working at the OIST for only approximately two years at the time of the interview, and although he was acquainted with most of the respondents, he often was unaware of the details of the episodes of medical examinations. Therefore, the occupational physician asked low-context questions, such as fact-checking and confirming feelings from a neutral perspective [24,25,26]. By consciously departing from the patient–medical provider relationship and reliving the foreigner’s trials and tribulations, the interviewees’ fears, hesitations, values, and “what they feel they need” were revealed as subthemes of the study. The interviewees were not aware that they harbored such feelings but rather had only an assertive, almost aggressive impression of the subjects when they encountered difficulties. The fact that the truth of the same episode differs based on different standpoints is often discussed from an elemental constructivist perspective [27,28,29]. However, this, once again, highlighted the need for such a qualitative study.

In theme (a), the interviewees spoke about similar content in different ways. In theme (b), where stories of difficult experiences were told, not only the attending physician and the patient but also many other parties such as receptionists, pharmacists, interpreters, colleagues, friends, family members, and health center staff appeared. The content of the experiences differed depending on the nature of the problem and the presence or absence of supporters such as family and friends.

In the trial-and-error process leading to the adaptation of theme (c), each interviewee experienced a story of transition from the viewpoint of difficulty and confusion to adaptation and confidence, with each interviewee having his or her own unique success story. It was not necessarily due to a single episode, nor was it a stereotypical black or white narrative development. There were differences in the process of adaptation between interviewees who had been in Japan for more than 20 years and those who had been in the country for less than a year. The intensity and earnestness of medical problems can have a significant impact on the scale of the timeframe of each individual’s story. Furthermore, support from family and friends during the medical visit is also thought to be a factor that significantly changes the development of the story. In addition, as the OIST has grown in size over time and medical facilities in the surrounding area have become more extensive, the transition from close face-to-face support to standardized support while maintaining a certain level of quality may also have been a factor for the development of narrative stories.

Although themes (d) and (e) are further stories told by the respondents who described their adaptation process in theme (c), what was said in theme (d) may have included their strength backed by their highbrow stance and their conscious or unconscious efforts to gain an advantage in unfavorable situations [30]. However, it should not be taken as a call for stereotypical consideration of one’s own culture or religion, which has been pointed out in previous studies, but rather as an appeal to treat each individual as a subject with his or her own personality and values.

Theme (e) was also discussed by an interviewee who had chronic illness and who was aware that the interviewers were a health center employees; they spoke candidly about their frustrations and demands that were not being adequately addressed, which seemed to indicate that the interviewees understood what was expected of them during the interview and tried to convey their feelings as faithfully as possible, being aware of their relational biases.

In the first half of the interview, the interviewees tended to deny any conflicts with Japanese medical personnel and excessively conveyed only the virtues of Japanese medicine. This may mean that the interviewees pandered to the interviewer or that they self-rationalized through efforts to adapt to the Japanese culture [31]. Therefore, at each break in the interview, the sub-interviewer asked neutral questions that were uncomfortable to ask, such as critical opinions about the medical profession and whether the interviewee had any opposing or ambivalent feelings about what they had answered. By doing so, we focused on neutrally accepting the interviewee’s critical and emotional opinions when they were expressed.

## 4. Discussion

In this study, we were mindful of the benefits and risks involved in interviewing health center staff involved in the respondents’ difficult medical examination experiences. At the OIST, where this study was conducted, a hierarchy was observed, in which university researchers play a leading role and the health center functions as a support service. Thus, there is less of an authority gradient in which medical personnel are in the upper echelon.

The interviewees may have had concerns about privacy or motivation to maintain a good relationship with the interviewers. As such, the two interviewers discussed their respective roles in advance and conducted interviews. The primary interviewer, who also participated in the story of the interviewee’s medical experience, acted as an empathetic listener and tried to elicit the interviewee’s inner feelings. To counter the possibility that the interviewees might take biased feelings and exaggerate or discard their stories with high expectations, the sub-interviewer tried to be a neutral and distanced listener. As a result, the inner difficulties and frank emotions associated with the foreigner’s medical visits, which could not be known in the everyday working relationship, and the adaptation process of each person were revealed.

It is possible that the interviewees objectively looked at their own situation, and while maintaining a metacognitive perspective that comprehends the impracticality of pushing their own culture and religion or demanding consideration in Japan, they expressed discomfort and anxiety in the form of frustration with insufficient explanations and convenience when they received medical care.

The highly skilled foreign residents we interviewed were generally satisfied with the level of healthcare in Japan and the commensurate cost burden. However, they also desired accessible and reliable healthcare, and language support was considered an essential component of healthcare. Unfortunately, access to healthcare for non-native Japanese speakers seems to be severely restricted in psychiatry and other fields, a situation that is particularly alarming. In addition, many seemed to feel alienated from information and support, not only in consultations with doctors, but also in the entire process of receiving healthcare.

Figure 1 organizes the relationship between the themes and subthemes over time, showing not only the specific adaptation process of the participants, but also the process of their internal efforts to overcome the gap between their ideals and reality in accepting the foreign culture. While (a) represents their ideal attitude toward a foreign culture they do not yet fully know, (b) represents their subsequent condemnation and bewilderment when they are confronted with it. This contradictory sense of (a) and (b) is integrated in the process of adaptation in (c), and in (d), they come to terms with Japanese medicine, fairly assess its strengths and weaknesses, and try to understand the doctor’s aloofness in the higher context and Japanese implicit communication style. Nevertheless, the demand for privacy rights and self-determination remained, as seen in the content of theme (e) concerning loneliness and workplace health checkups. Moreover, some believed that a reliable primary care physician could solve those problems. An additional stage of adaptation beyond this may be necessary, although the participants of the present study did not indicate such.

Many previous studies have examined how language barriers and low health literacy within the process of cross-cultural adaptation impact immigrant utilization of health services in other countries [32,33]. In Japan, many highly skilled foreign professionals have lifestyles that do not stick to their own cultural norms, as well as high health literacy and self-regulation skills. However, difficulties in communication, even in English, and limited access to information when receiving healthcare may overturn the assumption that they have self-regulation skills, reduce their sense of self-efficacy, and influence their behavior when receiving healthcare. Codes for the adjustment process among our participants indicated that support from family and friends mitigated the shock of encountering the difficulties with Japanese healthcare, and that chronically ill patients were anxious and frustrated by the lack of social support in the community. Supporting the maturation of mutual support systems among the expatriate community may also be important.

The present study did not identify any factors that might explain the difference between participants who were relatively adaptable and those who reported difficulties. However, the latter were more likely to prefer a primary care physician. Participants who reported a need for a primary care physician indicated that they wanted not only the convenience of a one-stop clinic but also a medical coordinator and emotional support. While it would be ideal if a regional cooperative system centered on primary care physicians could be established, the general shortage of medical practitioners capable of treating foreigners is thought to be a major barrier. The extent to which medical interpreters, Japanese international nurses [3], and medical coordinators [34] can fulfill the roles expected of primary care physicians, or how these roles can be shared, must be examined in the future.

Regarding language support, in addition to the assignment of medical interpreters, basic measures that can be introduced across all facilities, such as multilingualization of forms and use of bilingual tables, could be highly effective and helpful. On the other hand, some participants noted that in situations such as psychiatric consultations, where sensitive communication and strict protection of privacy are required, they would prefer doctors to communicate directly in a foreign language without the use of an interpreter. Individual attitudes toward the right to privacy and informed self-determination may also be influenced by cultural differences and norms therein. Given the wide range of ideals concerning privacy and one’s locus of control of health, it will be important to determine how to incorporate these diverse needs into actual healthcare for foreigners.

### Limitations

Because this study was conducted by occupational physicians and staff of the University Health Center, some participants may have refrained from being completely honest or making critical remarks about the medical staff. Still, interview content included candid statements reflecting distrust of medical examinations, suggesting that bias from the relationship between the researcher and participants was not strong. That said, the “co-construct” based on symmetrical relationships, which is important in narrative research, was probably sufficient in the present study. We believe we must further examine this point.

When interpreting the study findings, it will also be necessary to consider the regional nature of Okinawa and the specificity of an English-speaking university, acknowledging that generalizability may be limited by the fact that this is a qualitative study. Moreover, given that our participants are highly skilled foreign professionals, i.e., a group of individuals with both a high intellectual level and a high degree of economic stability, our findings would likely be markedly different if poor immigrants were targeted instead. In some areas of Japan, healthcare resources for foreigners are limited, and various problems with the healthcare system are inevitable. Future studies on this topic would benefit from expanding the scope of the survey in order to conduct a quantitative survey aimed to gauge a clear understanding of the needs of various foreign residents.

## 5. Conclusions

The present study, which employed a narrative-oriented approach, revealed that the target population (i.e., foreigners who are highly specialized in their fields) valued Japanese healthcare, but also felt socially vulnerable due to difficulties with communication and an inaccessibility of medical information. They also voiced the need for quality medical information, including appropriate and sufficient explanations, rather than individual cultural and religious considerations. This may have been influenced by the interviewees’ values of being adaptable as cosmopolitan citizens and being proud of their highbrow cross-cultural understanding. However, it was more likely an expression of their desire to be treated as an individual rather than as stereotypical cultural or religious categories based on their country of origin. Although the degree and process of adaptation to community medicine in Japan varied, the stories told by the respondents revealed insights that the listeners had not thought of during the course of their work, and it was apparent that they used frank expressions with an appropriate understanding of what was expected of them. In order to meet these needs, it is necessary to provide not only appropriate language support, but also medical knowledge, local medical coordination, and even emotional support as needed. Further consideration is required to determine how the Japanese healthcare system can fulfill such a multifaceted and comprehensive role, how appropriate healthcare can be offered to foreigners, and how these different responsibilities should be divided up and shouldered by involved parties. Collaboration between foreigners and Japanese medical professionals is necessary to address all of these issues.

## Figures and Tables

**Figure 1 healthcare-10-01694-f001:**
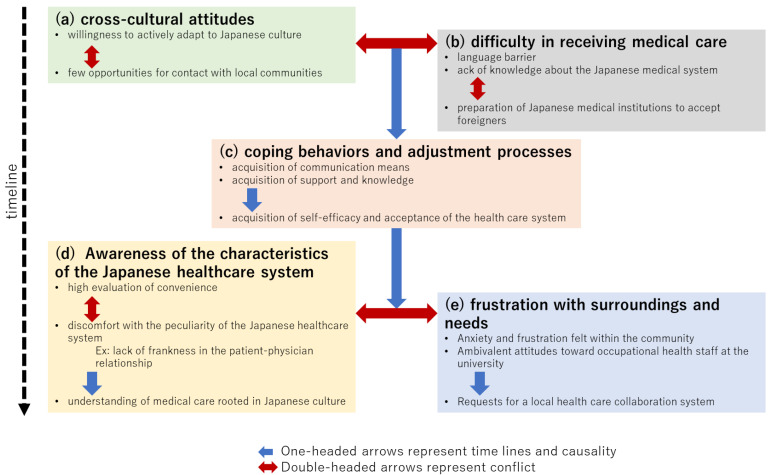
Overview of themes and subthemes extracted from participant experiences with healthcare in Japan. Double-headed arrows indicate conflict; single-headed arrows indicate causality.

**Table 1 healthcare-10-01694-t001:** Participant attributes.

**Participant No.**	1	2	3	4	5	6	7
**Age group (years)**	40–50	30–40	40–50	20–30	30–40	30–40	40–50
**Sex**	Male	Male	Male	Male	Male	Female	Female
**Occupation/Educational status**	Faculty /Staff	Faculty /Staff	Faculty /Staff	Student, Herein-period government administration	Student, Herein-period government administration	Faculty /Staff	Faculty /Staff
**Geographical area of origin**	South Asia	Europe	South Africa * 1	North America * 1	Europe	North America * 1	North America * 1
**Number of years since coming to Japan (years)**	20–25	5–10	5–10	<5	<1	<5	<5
**Japanese language proficiency**	Fluent (language skill)	Ordinary (daily life) conversation	Ordinary (daily life) conversation	Ordinary (daily life) conversation	Greeting phrase only	Fluent (language skill)	Greeting phrase only
**Number of codes created**	26	33	26	34	24	25	53
**Participant No.**	8	9	10	11	12	13	
**Age group (years)**	30–40	30–40	30–40	40–50	50–60	30–40	
**Sex**	Female	Female	Female	Female	Male	Female	
**Occupation/Educational status**	Student, Herein-period government administration	Faculty /Staff	Student, Herein-period government administration	Faculty /Staff	Faculty /Staff	Faculty /Staff	
**Geographical area of origin**	Europe	North America * 1	Near/Middle East	East Asia	Europe	North America * 1	
**Number of years since coming to Japan (years)**	<5	5–10	5–10	5–10	5–10	5–10	
**Japanese language proficiency**	Greeting phrase only	Ordinary (daily life) conversation	Ordinary (daily life) conversation	Ordinary (daily life) conversation	Ordinary (daily life) conversation	Ordinary (daily life) conversation	
**Number of codes created**	32	29	38	23	57	34	

* Native English speaker.

**Table 2 healthcare-10-01694-t002:** Interview guide for the semi-structured interview to examine healthcare issues for foreign residents from the perspective of foreign residents.

■ Setting Place: B251, face-to-face or Zoom meeting. One hour duration, interview content recorded, researcher will conduct the interview, and gift voucher for 1000 yen will be presented as compensation (after the interview). ■ Introduction Self-introduction, explanation that the interviewer’s position is completely neutral, and informant can ask questions and confirm during the interview. ■ Explanation of research study and healthcare issues among foreigners With the increase in the number of foreigners visiting Japan, urgent medical issues regarding foreigners as perceived by Japanese healthcare providers generally include the lack of medical interpreters, unpaid medical expenses, and the introduction of infectious diseases into the country. ■ Questions about basic interviewee attributes Age, sex, years since coming to Japan, level of fluency in Japanese, family structure (spouse, child), and other special notes. ■ Type of healthcare experienced by interviewees Acute care, chronic care, outpatient, hospitalization, health checkup, vaccinations, maternal and child health, etc. ■ Evaluation of the healthcare in Japan Pros and cons. ■ Evaluation of the healthcare in the interviewee’s country of origin Pros and cons. ■ Anecdotal experiences Specific difficult experiences. Specific good experiences. ■ Hearing individual perspectives of interviewees ▶ Confidentiality and self-determination. ▶ Opinions on communication with medical staff, healthcare systems, and accessibility. ▶ Opinions on the balance between collective and individual interests from a public health perspective. ▶ Resource (resource person), role of community. ■ Review and summary, additional questions

**Table 3 healthcare-10-01694-t003:** Difficulties with receiving medical examinations in Japan as perceived by participants.

Theme	Subtheme	Code
Cultural adaptation	Willingness to adapt actively to Japanese culture	Abundant overseas experience as a high-level foreign human resource, and an international sense that enables them to relativize their own culture
		Cultural criticism
		Japanese character
		Willingness to learn Japanese
		Familiarity with the Japanese people and Japanese culture
		Unexpected difficulties of living in a real Japanese language environment
	Little contact with the community	Generous support from the university in terms of daily life
Difficulties experienced in order to see a doctor	Language barrier	In medical institutions where patients are expected to speak Japanese and staff other than doctors do not speak English, so foreigners are at a loss at the hospital reception desk in the first place
		Filling out medical forms written only in Japanese is difficult, and there is little multilingual support
		It is unrealistic to expect foreigners to acquire a level of Japanese that will allow them to explain and understand healthcare
		Especially for foreigners, it is difficult to communicate with medical personnel in case of sudden or serious illness
		Differences in perception between foreigners who believe that the responsibility for language support lies with the medical institution and Japanese medical institutions that do not provide language support
	Lack of knowledge about the Japanese healthcare system	In Japan, medical departments are subdivided, and medical institutions range from large hospitals to small clinics, making it difficult to know which medical institution to go to first
		Difficulties understanding the flow and payment procedures at general hospitals, from reception to payment and prescriptions
	Readiness to accept	Experiences being told by psychiatrists and obstetricians that they cannot be treated because they are foreigners
		Experiences being required by medical institutions to bring their own interpreter in the event of a sudden illness or serious situation
Coping behaviors and adaptive processes	Individual participants’ narrative which is awkward especially on information	Family members and acquaintances are expected to accompany the patients to the hospital and assist them, allowing them to survive the emergency
		The costs, risks, and benefits of hiring a professional or volunteer interpreter
		Lack of English-language websites and distrust due to likely outdated content
		While ICT equipment is inexpensive and easy to use, its applications are limited due to its low accuracy
		Recognize the practical effectiveness of trying to communicate openly in both Japanese and English, even if imperfectly
	Support and knowledge	Learn how to obtain healthcare from familiar sources, e.g., gather information from acquaintances, colleagues, and university health centers
		Experience the convenience of seeing a doctor at a medical institution with full-time interpreters and other language support
		Familiarize oneself with the procedures and learn how to go to the hospital alone at non-emergent times
	Achievement of self-efficacy and satisfaction with the healthcare system	By giving up or compromising, they feel a sense of security that they can manage to solve the problem and come to terms with the healthcare provided
Perceptions of the characteristics of the Japanese healthcare system	High ratings for convenience	Positive evaluation of inexpensive and uniform medical costs
		Favorable evaluation of easy access to advanced medical equipment such as CTs and MRIs, as well as access to medical specialists
		Confidence that Japanese healthcare meets a certain standard regardless of the medical institution
		Positive evaluation of not feeling discrimination against foreigners by healthcare workers
	Discomfort with the peculiarities of the Japanese healthcare system	Discomfort with the Japanese healthcare system and feeling that it is inconvenient relative to that in one’s own country of origin
		Complexity of paperwork, bureaucratic inflexibility regarding procedures
	Direct and open relationships	Difficulty in establishing a direct and open relationship between patients and physicians because of the still-prevalent stereotype of paternalism wherein physicians explain and patients follow their instructions without questioning
	Foreigners’ understanding of healthcare	An understanding and acceptance of the advantages, disadvantages, and inconveniences of Japanese medicine as a characteristic of Japanese culture
Frustration	Anxiety and frustration felt in the community	The loneliness of living in a foreign land, far from home and family
		Loneliness of not having someone close by to share the experience of illness or to talk to
		Stigma due to fear of invasion of privacy and lack of understanding of surroundings
	Ambivalent attitudes toward occupational health staff at universities	Trust in occupational health staff as a source of information about the Japanese healthcare system and distrust of workplaces that force employees to undergo health checkups and keep track of health information
	Requests for a regional medical cooperation system	The need for reliable primary care physicians who can address everyday problems of foreign patients and serve as a base for medical coordination

**Table 4 healthcare-10-01694-t004:** Types of language support mentioned by participants and their advantages and disadvantages.

Type of Support	Features and Advantages/Disadvantages
**Human interpreter**	The individual skills of medical interpreters vary, they are difficult to secure at the last minute, they are expensive, they help with various hospital procedures, and they should be readily available at least in public hospitals and larger hospitals.
**Translation apps and online interpreters**	Translation apps are inexpensive (free) and easy to use, but lack accuracy and reliability and are limited to ancillary use. Online medical interpreters are inexpensive, reliable, and do not require an appointment. Online interpretation services provided by local governments are not well known and are rarely used in hospitals.
**Information about the website and multilingual documentation**	The English versions of hospital and government websites are often inaccurate because their content is more limited or outdated relative to the Japanese version. Multilingual documents in bilingual (English and Japanese) are very useful but not always available.
**Medical professionals with foreign language skills**	In psychiatric care, direct and accurate communication is critical. When choosing a medical institution, care options are often dependent on whether or not the doctor speaks English, and not by the doctor’s medical skills. There is a serious shortage of English-speaking physicians.

## Data Availability

Not applicable.

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
