# Peer review of "Narratives on the Current Medical Situation in Japan According to Highly Specialized Foreign Professionals"

_healthcare, 2022, doi:10.3390/healthcare10091694_

Round 1
Reviewer 1 Report
The article seems a bit descriptive in the empiriccal paragraph and the conclusion. The author(s) should better interpretate and analyse the data and the interview, as now they are just quoted without critical analysis and interpretation.
Conclusion must be improved with some more theoretical reflections.
Author Response
Reviewer 1 Open Review
The article seems a bit descriptive in the empirical paragraph and the conclusion. The author(s) should better interpretate and analyse the data and the interview, as now they are just quoted without critical analysis and interpretation.
Conclusion must be improved with some more theoretical reflections.
Response: Thank you for your insightful comment and for reviewing our manuscript. We have now added our analysis and interpretation of the results and conclusions as a narrative as mentioned below.
First, we added the following analysis of the results from a narrative perspective.
At the beginning of this study, the interviewers anticipated that the content that interviewees considered to be difficult at the time of their visit, and their demands regarding respect for culture and religion would be largely determined by their cultural backgrounds. However, the interviewees' experiences could be analyzed in terms of length of stay and stage of experience and organized as themes (a)–(e) of the story of the adjustment process shared by the interviewees. In the following sections, we further describe the relationship between the interviewee and interviewer, including the aspects of the individual narratives along the time axis that link each theme.
The main interviewers, public health nurses, and administrative staff had been engaged in supporting researchers for about 10 years, since the opening of the OIST. Public health nurses, in particular, had direct contact with patients, so they remembered most episodes of medical visits described by the interviewees from their own perspective. For the interviewees, sharing episodic experiences with the interviewer seemed to evoke a sense of wanting to be heard and understood, which in this study worked to their advantage in understanding their trials and tribulations. In cases where the health worker was a friend or family member of the interviewee, the administrative staff acted as the main interviewer to avoid any bias. The sub-interviewee, an industrial physician, had been working at the OIST for only approximately two years at the time of the interview; and although s/he was acquainted with most of the respondents, they often were unaware of the details of the episodes of medical examinations. Therefore, occupational physicians asked low-context questions, such as fact-checking and confirming feelings from a neutral perspective [24–26]. By consciously departing from the patient-medical provider relationship and reliving the foreigner's trials and tribulations, the interviewees' fears, hesitations, values, and “what they feel they need” were revealed as subthemes of the study. The interviewees were not aware that they harbored such feelings but rather had only an assertive, almost aggressive impression of the subjects when they encountered difficulties. The fact that the truth of the same episode differs based on different standpoints, is often discussed from an elemental constructivist perspective [27–29]. However, this, once again, highlighted the need for such a qualitative study.
In theme (a), the interviewees spoke about similar content in different ways. In theme (b), where stories of difficult experiences were told, not only the attending physician and the patient but also many other parties such as receptionists, pharmacists, interpreters, colleagues, friends, family members, and health center staff appeared. The content of the experiences differed depending on the nature of the problem and the presence or absence of supporters such as family and friends.
In the trial-and-error process leading to the adaptation of theme (c), each interviewee experienced a story of transition from the viewpoint of difficulty and confusion to adaptation and confidence, with each interviewee having his or her own unique success story. It was not necessarily due to a single episode, nor was it a stereotypical black or white narrative development. There were differences in the process of adaptation between interviewees who had been in Japan for more than 20 years and those who had been in the country for less than a year. The intensity and earnestness of medical problems can have a significant impact on the scale of the timeframe of each individual’s story. Furthermore, support from family and friends during the medical visit is also thought to be a factor that significantly changes the development of the story. Furthermore, as the OIST has grown in size over time and medical facilities in the surrounding area have become more extensive, the transition from close face-to-face support to standardized support while maintaining a certain level of quality may also have been a factor for the development of narrative stories.
Although themes (d) and (e) are further stories told by the respondents who described their adaptation process in theme (c), what was said in theme (d) may have included their strength backed by their highbrow stance and their conscious or unconscious efforts to gain an advantage in unfavorable situations [30]. However, it should not be taken as a call for stereotypical consideration of one's own culture or religion, which has been pointed out in previous studies but rather as an appeal to treat each individual as a subject with his or her own personality and values.
Theme (e) was also discussed by an interviewee who was in the hospital for a chronic illness and who was aware that the interviewer was a health center employee, spoke candidly about her frustrations and demands that were not being adequately addressed, which seemed to indicate that the interviewees understood what was expected of them during the interview and tried to convey their feelings as faithfully as possible, being aware of their relational biases.
In the first half of the interview, the interviewees tended to deny any conflicts with Japanese medical personnel and excessively convey only the virtues of Japanese medicine. This may mean that the interviewees pandered to the interviewer or that they self-rationalized through efforts to adapt to the Japanese culture [31]. Therefore, at each break in the interview, the sub-interviewer asked neutral questions that were uncomfortable to ask, such as critical opinions about the medical profession and whether the interviewee had any opposing or ambivalent feelings about what they had answered. By doing so, we focused on neutrally accepting the interviewee's critical and emotional opinions when they were expressed.
Furthermore, we improved our conclusions based on the above discussion. The following text has been added.
This may have been influenced by the interviewees' values of being adaptable as cosmopolitan citizens and being proud of their highbrow cross-cultural understanding. However, it was more likely an expression of their desire to treat everyone as an individual rather than as stereotypical cultural or religious categories based on their country of origin. Although the degree and process of adaptation to community medicine in Japan varied, the stories told by the respondents revealed insights that the listeners had not thought of during the course of their work, and it was apparent that they used frank expressions with an appropriate understanding of what was expected of them.

Reviewer 2 Report
Dear Authors,
Thank you for the opportunity to read your manuscript.
It raises important intercultural issues, especially those related to communication. Being able to explore the views and feelings of culturally different people, as it were, 'first-hand' is an important contribution to learning about another culture.
Despite the high value of the manuscript and the study, I have a few doubts:
1 - the introduction is a bit lapidary. It would have been good to write some characteristics of Japanese culture that may not be understood by culturally different people
2 - the purpose of the study is missing
3 - the participants of the study should be separated from the description of the research method
4 - the characteristics of the study participants (Table 2) would be better moved to the section - participants
5 - were the people who conducted the interviews trained? This is important especially in the context of the last part of the interview: Review and summary, additional questions
6 - did respondents know they would receive a gift voucher after the interview?
7 - phrases such as: We were also not well convinced of the cultural and religious considerations and stereotypes of Japanese medical staff (lines 130-131) should not be in the description of the study but rather in the limitations of the study
8 - 14 literature items? The authors seem to have explored the issues poorly.
Author Response
Reviewer 2 Open Review
1 - the introduction is a bit lapidary. It would have been good to write some characteristics of Japanese culture that may not be understood by culturally different people
Response 1: Thank you for your comment. We have now added the following background information, including the characteristics of Japanese culture.
In-depth research on the adaptation process of ethnic groups to different cultures has been conducted focusing on minorities, especially in the Western world, where ethnic migration has been historically repeated [4–8]. These studies have established a field of cross-cultural adaptation and understanding in healthcare settings [9–15]. However, Japan is an island nation comprising a single ethnic group, speaking almost exclusively Japanese. Although English education has long since become compulsory, the country has not yet become multi-ethnic or multilingual. In recent years, Japan has seen rapid globalization, highlighting the need to secure an efficient and young workforce to overcome a super-aging society. The Japanese government has commenced a policy of emphasizing the importance of tourism industry and attracting white-collar workers along with blue-collar workers. Consequently, the number of foreign tourists and foreign residents in Japan has been increasing rapidly. However, the system for accepting foreign nationals is still in its nascent stage; and the medical field has limited experience in accepting foreign personnel in various sectors. In addition, there is still little research from the perspective of cross-cultural understanding and response in this regard.
2 - the purpose of the study is missing
Response 2: Thank you for your comment. We have now added the following background to the purpose of the narrative interviews.
The OIST Health Center staff interviewed for this study provided medical consultation support to foreign resident faculty, staff, students, and accompanying family members since the opening of the university. Although we have dealt with many problems and complaints from both foreigners and Japanese healthcare providers regarding medical visits, these problems required merely practical and technical solutions. Until now, there has been no opportunity to deeply investigate or consider the cross-cultural communication and understanding that lies behind problems in daily work. Therefore, this study aimed to extract stories, including personal feelings and opinions regarding foreigners’ perception of difficulties in visiting Japanese medical facilities and their experiences of overcoming these difficulties. The OIST health center staff, who had been involved in supporting foreign residents in their medical visits, listened to their stories and tried to understand them. The outcome of the study was to gain a foothold for the inclusion of foreigners in local healthcare and medical support professions in Japan.
3 - the participants of the study should be separated from the description of the research method
Response 3: Thank you for your comment. We have separated the descriptions as per your comment.
4 - the characteristics of the study participants (Table 2) would be better moved to the section - participants
Response 4: Thank you for your comment. We have now moved the description of participant characteristics.
5 - were the people who conducted the interviews trained? This is important especially in the context of the last part of the interview: Review and summary, additional questions
Response 5: Thank you very much for your comment. We believe that the question we received described the role and background of the interviewer in the interview. We have now added the following statement to our results.
The main interviewers, public health nurses, and administrative staff had been engaged in supporting researchers for about 10 years, since the opening of the OIST. Public health nurses, in particular, had direct contact with patients, so they remembered most episodes of medical visits described by the interviewees from their own perspective. For the interviewees, sharing episodic experiences with the interviewer seemed to evoke a sense of wanting to be heard and understood, which in this study worked to their advantage in understanding their trials and tribulations. In cases where the health worker was a friend or family member of the interviewee, the administrative staff acted as the main interviewer to avoid any bias. The sub-interviewee, an industrial physician, had been working at the OIST for only approximately two years at the time of the interview; and although s/he was acquainted with most of the respondents, they often were unaware of the details of the episodes of medical examinations. Therefore, occupational physicians asked low-context questions, such as fact-checking and confirming feelings from a neutral perspective [24–26]. By consciously departing from the patient-medical provider relationship and reliving the foreigner's trials and tribulations, the interviewees' fears, hesitations, values, and “what they feel they need” were revealed as subthemes of the study. The interviewees were not aware that they harbored such feelings but rather had only an assertive, almost aggressive impression of the subjects when they encountered difficulties. The fact that the truth of the same episode differs based on different standpoints, is often discussed from an elemental constructivist perspective [27–29]. However, this, once again, highlighted the need for such a qualitative study.
Furthermore, the following was added to the discussion.
In this study, we were mindful of the benefits and risks involved in interviewing health center staff involved in the respondents' difficult medical examination experiences. At the OIST, where this study was conducted, a hierarchy was observed, in which university researchers play a leading role and the health center functions as a support service. Thus, there is less of an authority gradient in which medical personnel are in the upper echelon.
The interviewees may have had concerns about privacy or motivation to maintain a good relationship with the interviewers. As such, the two interviewers discussed their respective roles in advance and conducted interviews. The primary interviewer, who also participated in the story of the interviewee's medical experience, acted as an empathetic listener and tried to elicit the interviewee's inner feelings. To counter the possibility that the interviewees might take critical feelings and exaggerate or discard their stories with high expectations, the sub-interviewer tried to be a neutral and distanced listener. As a result, the inner difficulties and frank emotions associated with the foreigner's medical visits, which could not be known in the everyday working relationship, and the adaptation process of each person were revealed.
It is possible that the interviewees objectively looked at their own situation, and while maintaining a metacognitive perspective that comprehends the impracticality of pushing their own culture and religion or demanding consideration in Japan, they expressed discomfort and anxiety in the form of frustration with insufficient explanations and convenience when they received medical care.
6 - did respondents know they would receive a gift voucher after the interview?
Response 6: Thank you for your comment. Yes, the interviewees were aware of this. We have added the following statement to the text to clarify it:
The amount given as a lost opportunity cost for an hour’s interview was 1,000 yen (under 10 US dollars), which is roughly equivalent to the hourly minimum wage in Japan. The researcher intended that the gift card would show appreciation for participation but not be a substantial incentive enough to participate.
7 - phrases such as: We were also not well convinced of the cultural and religious considerations and stereotypes of Japanese medical staff (lines 130-131) should not be in the description of the study but rather in the limitations of the study
Response 7: Thank you for your comment. This part of the text was inappropriate; therefore, please note, we have now deleted it.
8 - 14 literature items? The authors seem to have explored the issues poorly.
Response 8: Thank you for your comment. In the Background section, we referred to a paper on cross-cultural adaptation abroad. We have also added prior literature on narrative approaches to the section which was added in the Results and Discussion section.

Round 2
Reviewer 1 Report
Fine